# Analyses of Physical and Chemical Compositions of Different Medicinal Specifications of CRPV by Use of Multiple Instrumental Techniques Combined with Multivariate Statistical Analysis

**DOI:** 10.3390/molecules27103285

**Published:** 2022-05-20

**Authors:** Yang Wang, Weiliang Cui, Guowei Pang, Lewen Xiong, Qingzhi Liu, Lili Xu, Huifen Li, Yongqiang Lin

**Affiliations:** 1School of Pharmacy, Shandong University of Traditional Chinese Medicine, Jinan 250355, China; wangyang961125@163.com (Y.W.); 15866682186@163.com (G.P.); lewenxiong@126.com (L.X.); qingzhi0905@163.com (Q.L.); hshpipi@163.com (L.X.); 2Shandong Institute for Food and Drug Control, Jinan 250101, China; xinyao2002@126.com; 3State Key Laboratory of Generic Manufacture Technology of Chinese Traditional Medicine, Linyi 276005, China

**Keywords:** Geqingpi, Sihuaqingpi, *Citrus reticulata* Blanco, Citri Reticulatae Pericarpium Viride, flavonoid glycosides, flavonoid aglycones, polymethoxyflavones, TCM, UHPLC-Q-Exactive Orbitrap-MS, FT-IR

## Abstract

Citri Reticulatae Pericarpium Viride (CRPV) is the processed product of *Citrus reticulata* Blanco. We systematically analyzed two CRPV types, Geqingpi (GQP) and Sihuaqingpi (SHQP), based on powder color, microscopic characteristics, and chemical composition. In addition, we characterized their constituents via ultra-high-performance liquid chromatography with hybrid quadrupole-orbitrap mass spectrometry (UHPLC-Q-Exactive Orbitrap-MS). Both showed significant differences in their powder color and microscopic characteristics. Fourier-transform infrared (FT-IR) spectroscopic analysis results showed that the C=O peak absorption of carboxylic acids and their carbonyl esters in SHQP was higher than that of GQP, while the C-OH and C-H plane bending peaks of polysaccharides were lower than those of GQP. We analyzed these data via similarity analysis, PCA, and OPLS-DA. GQP and SHQP had large distinct differences. Based on the mass measurements for molecular and characteristic fragment ions, we identified 44 main constituents from CRPV, including different flavonoid glycosides and flavonoid aglycones in SHQP and GQP, respectively. We found luteolin-6-C-glucoside, orientin, rhoifolin, and pilloin solely in SHQP, and naringenin and hesperetin only in GQP. The peak area measurements showed GQP having a higher flavonoid glycoside (narirutin, hesperidin, etc.) content, whereas SHQP had a higher polymethoxyflavone (nobiletin, tangeretin, etc.) content. Since we holistically analyzed two CRPV types, the results can not only support future pharmacological research, but also provide a scientific basis for formulating more reasonable CRPV quality standards and guide its clinical potential as a precision medicine.

## 1. Introduction

CRPV is a traditional Chinese medicine (TCM) commonly used in China. According to the “Pharmacopoeia of the People’s Republic of China”, CRPV is the processed product of the peel of the dried young or immature fruit of *Citrus reticulata* Blanco and its cultivars. The earliest records of its medicinal application can be traced back to the Tang Dynasty in China [1]. As per the TCM theory, CRPV soothes the liver and breaks Qi, eliminates accumulation, and resolves stagnation [2]. According to its harvesting time, CRPV can be divided into two types: GQP and SHQP. GQP is the young citrus fruit and is often collected from May to June; while SHQP, often collected from July to August, is the peel of the immature citrus fruit. When used, GQP is directly sliced, whereas SHQP is longitudinally cut into four pieces towards the base of the peel, and all the flesh pieces are removed. According to ancient Chinese herbal books, statements like “broken head-like lotus petals [3]”, “cross out the flesh and divided into four pieces called lotus-like CRPV [4]”, and “it is better to use a knife to cut the lotus petals on the CRPV”, indicate that SHQP was used in ancient China [5]. However, GQP is currently often used in the Chinese mainland. Since CRPV used in ancient and modern China has different types, the difference in composition and efficacy between the two types is worth exploring.

At present, SHQP costs twice that of GQP in the Chinese mainland. GQP, which is mainly used as the raw material of decoctions, has a market share of 80–90%. SHQP is mostly used for preparing Chinese patent medicine, and is also sold to South Korea, Hong Kong, etc. This shows that there are price and grade differences in the current commodity circulation types of CRPV. It is necessary to find the different components of the two types, to help establish the CRPV grade standards and provide a reference for the identification method of Chinese patent medicines with different CRPV.

Modern studies have found that CRPV inhibits (a) gastrointestinal smooth muscle movement [6], (b) duodenum contraction [7], (c) the movement of the longitudinal muscle strips of the small intestine, (d) the contraction of the longitudinal muscle of the ileum, and stimulating the smooth muscle of the bladder [8,9,10]; it also affects the bioelectrical activity of the gastrointestinal tract. It can also reduce the amplitude of the gastric electric slow-wave and prolongs the period [11].

Clinical studies have found that CRPV extract can induce apoptosis in the human colon cancer cell line SNU-C4, thus suggesting that CRPV has anticancer effects [12]. Polymethoxyflavones (PMFs) in CRPV improve metabolic disorders like hyperlipidemia, which is induced by a high-fat diet by modulating the gut microbiome and amino acid metabolism [13,14]. It also inhibits the proliferation of gastric cancer cells by inducing gastric cancer cell death via upregulation of the RARβ protein [15]. Bioactive flavonoids (narirutin, nobiletin, tangeretin, etc.) can protect the liver, reduce its damage, and prevent liver diseases [16,17]. Nobiletin may inhibit the expression of IL-6, TNF, and CCL2, which have anti-inflammatory and antioxidant properties. It can relieve the clinical symptoms of colitis mice and reduce their liver and kidney damage [18,19,20]. Tangeretin may inhibit the expression of interleukin-23 and other related proteins via the Notch signaling pathway, thereby treating acute lung injury [21]. At the same time, it may also inhibit the proliferation of liver cancer cells [22]. Narirutin has also exhibited anti-inflammatory and antioxidant activities, by activating the NF-κB and MAPKs pathways, and inhibiting various pro-inflammatory mediators like NO and PGE2 in macrophages [23]. It not only has a potential therapeutic effect on Alzheimer’s disease, but may also reduce alcohol-induced liver damage [24,25]. Hesperidin has not only reduced in vivo lipid accumulation, indicating its potential anti-hyperlipidemic effects, but also has anti-osteoporotic effects [26,27].

Although some studies have shown that GQP and SHQP are mostly similar in their compositions, the contents of flavonoids (hesperidin, narirutin, nobiletin, tangeretin, etc.) and alkaloids (synephrine, N-methyltyraminehydrochloride) are significantly different. While GQP has higher narirutin and hesperidin contents than SHQP, it has comparatively lower contents of nobiletin and tangeretin [28,29,30]. Understanding the pharmacodynamic material basis of the two CRPV types would help in guiding their rational clinical use.

In this study, we used colorimeter, microscope, FT-IR, and UHPLC-Q-Exactive Orbitrap-MS technology to comprehensively analyze the similarities and differences between GQP and SHQP based on powder color, microscopic characteristics, and chemical composition, to provide a scientific basis for formulating more reasonable CRPV quality standards and guiding its clinical development as a precision medicine.

## 2. Results

### 2.1. Powder Color Analysis

Upon analyzing the pictures of both GQP and SHQP powder, we found the GQP powder color was darker than that of SHQP (Figure 1). Based on the L*, a*, and b* values of 22 samples shown in Table 1, we can see that SHQP had a higher brightness L* value than GQP, thereby indicating that the SHQP powder color was brighter. Although the red-green degree value a*: GQP > SHQP > 0, indicated that both their powder colors were red, the SHQP color was redder. Furthermore, the yellow-blue degree value b*: SHQP > GQP > 0, indicated that both their powder colors were yellow, but the color of the SHQP was yellower.

### 2.2. Microscopic Cell Analysis

According to the microscopic observation, the color of the microscopic cells of GQP was darker, while that of SHQP was lighter, thus being consistent with the powder color results. According to the Chinese Pharmacopoeia 2020 edition, while the GQP microcells comprise calcium oxalate crystallization, epidermal cells of the pulp capsule, and hesperidin crystal (a), those of SHQP mainly comprise tracheal, calcium oxalate crystallization, exocarp, mesocarp parenchyma, hesperidin crystal, and stoma (b). The result is shown in Figure 2. However, while observing, we found that the characteristic cells of SHQP also existed in individual GQP batches.

### 2.3. FT-IR Analysis

The FT-IR spectrum of CRPV is shown in Figure 3, and the location of the most relevant features of the CRPV are shown in Table 2. We divided the infrared spectral absorption peaks of GQP and SHQP into five sections: the first was 3450–3350 cm^−1^; the second was 3000–2800 cm^−1^; the third was 1800–1350 cm^−1^; the fourth was 1300–1000 cm^−1^; and the fifth section was 900–600 cm^−1^. There were no significant differences between GQP and SHQP in the first and second sections. Affected by volatile oils, GQP and SHQP showed characteristic peaks near 3419 cm^−1^, which were broad and strong, resulting from the hydrogen bonding of free O-H. Due to flavonoids, GQP and SHQP showed characteristic peaks near 2923 cm^−1^, generated by the asymmetric stretching vibration of methylene C-H. There was a significant difference between GQP and SHQP in the third sections, affected by carboxylic acids and their esters. GQP and SHQP showed characteristic peaks near 1746 cm^−1^, which were carbonyl C=O stretching vibration peaks, with the absorption peak of SHQP in this band being more noticeable than that of GQP, leading to speculation that the content of carboxylic acids and its esters in SHQP was higher than that in GQP. GQP and SHQP showed characteristic peaks near 1639 cm^−1^, which were C=C or aromatic ring skeleton vibration superimposed peaks. However, GQP showed two absorption peaks at 1646 cm^−1^ and 1609 cm^−1^, due to asymmetric stretching vibrations. In the fourth and fifth sections, the absorption peaks of GQP were more noticeable than those of SHQP, with the fourth section being 1300–1000 cm^−1^, due mostly to C-O stretching vibration peaks, while the C-OH stretching vibration peaks of polysaccharides were mostly around 1070 cm^−1^. The fifth segment peak of 900–600 cm^−1^ was mostly the vibration absorption peak of carbohydrates, affected by polysaccharides. GQP and SHQP showed characteristic peaks around 800 cm^−1^, which were C-H plane bending vibration absorption peaks. We speculated that GQP had a higher polysaccharide content than SHQP.

### 2.4. Qualitative Analysis of Constituents

We analyzed the CRPV samples using UHPLC-Q-Exactive Orbitrap-MS in both positive and negative ion modes (Figure 4). Considering the chromatographic peaks, the MS spectra obtained in positive ion mode were better than those obtained in negative ion mode. Most of the PMFs showed abundant peaks in the positive ion mode rather than in negative ion mode. Details of the identified compounds are presented in Table 3. Finally, 44 major components were identified or preliminarily identified, including 5 flavone-C-glycosides, 4 flavone-O-glycosides, 4 flavanone-O-glycosides, 4 flavanone aglycones, 21 PMFs, 3 alkaloids, 2 limonoids, and 1 other compound (Table 3). Chemical structures of the major constituents are shown in Figure 5, Figure 6 and Figure 7 and Table 4 and Table 5. Among them, luteolin-6-C-glucoside, orientin, rhoifolin, and pilloin were unique to SHQP, most of which were flavonoid glycosides. However, flavanone aglycones like naringenin and hesperetin were unique to GQP.

#### 2.4.1. Analysis of Flavonoids

In this study, we identified 44 major components including 5 flavone-C-glycosides, 4 flavone-O-glycosides, 4 flavanone-O-glycosides, 4 flavanone aglycones, 21 PMFs, 3 alkaloids, 2 limonoids, and 1 other compound from the CRPV samples. Consistent with the previously reported results [31], the retro Diels–Alder (rDA) reaction often exists in the cracking process of most flavonoids.

##### Analysis of Flavonoid-O-Glycosides

Flavonoid-O-glycosides are common flavonoid compounds commonly found in citrus fruits, and are linked to rutinose or neohesperidose via the C-7 hydroxyl group, which links the disaccharide to the aglycone [32,33]. Based on previous reports [34,35,36,37], we identified and tentatively characterized four flavone-O-glycosides (compounds **8**, **10**, **14**, and **15**) and four flavanone-O-glycosides (compounds **9**, **12**, **16**, and **20**). For instance, compared with reference standards, we unambiguously identified compounds **9** and **12** as narirutin and hesperidin, respectively. Hesperidin showed a quasi-molecular ion [M + H]^+^ at *m*/*z* of 611.19501 (C_28_H_34_O_15_), due to the continuous loss of a glucose (162 Da) and a rhamnose (146 Da), thereby giving characteristic product ions at *m*/*z* of 303.08820. Narirutin gave an [M + H]^+^ signal at *m*/*z* of 581.08439 (C_2__7_H_3__2_O_1__4_), and its fragmentation pattern shared similarity with hesperidin, which generated MS/MS ions at *m*/*z* of 273.07533 [M + H-Glc-Rha]^+^.

##### Analysis of Flavonoid-C-Glycosides

Fragmentations in most flavonoid-C-glycosides were based on the aglycones of flavones [38]. In the MS/MS spectra, the main product ion of flavone-C-glycosides usually occurred in the glycosyl moiety, which was generated by the loss of water molecule(s) and the glycosidic methylol group as formaldehyde [39]. According to previous reports [34,35,36,37], the product ions [M + H–n18Da]^+^ are diagnostic ions, while the other two fragments, [M–H–90Da]^−^ and [M–H–120Da]^−^, revealed that a C-glycoside was connected with hexose units. In the present study, we tentatively identified five flavone-C-glycosides (compounds **3**, **4**, **5**, **6**, and **7**) in CRPV, in which compounds **6** and **7** belong to SHQP. For example, compound **3** showed a quasi-molecular ion [M + H]^+^ at *m*/*z* of 611.15961 (C_27_H_30_O_16_), which generated MS/MS ions at *m*/*z* of 557.12787 [M + H-OCH_3_-H_2_O]^+^, 473.10663 [M + H-C_4_H_8_O_4_-H_2_O]^+^, and 353.06503 [M + H-2C_4_H_8_O_4_-H_2_O]^+^. Compound **4** exhibited a quasi-molecular ion [M + H]^+^ at *m*/*z* of 595.16406 (C_27_H_30_O_15_), which generated MS/MS ions at 457.11102 [M + H-C_4_H_8_O_4_-H_2_O]^+^. Compound **5** displayed a quasi-molecular ion [M + H]^+^ at *m*/*z* of 625.17480 (C_28_H_22_O_16_), which generated MS/MS ions at *m*/*z* of 487.11911 [M + H-C_4_H_8_O_4_-H_2_O]^+^ and 367.08112 [M + H-2C_4_H_8_O_4_-H_2_O]^+^. Compound **6** exhibited a quasi-molecular ion [M + H]^+^ at *m*/*z* of 449.10727 (C_21_H_20_O_11_), which generated MS/MS ions at 413.08545 [M + H-2H_2_O]^+^ and 395.07538 [M + H-3H_2_O]^+^. Compound **7** presented a quasi-molecular ion [M + H]^+^ at *m*/*z* of 449.10727 (C_21_H_20_O_11_), which generated MS/MS ions at *m*/*z* of 413.08545 [M + H-2H_2_O]^+^ and 395.07538 [M + H-3H_2_O]^+^. On comparing with the flavonoid-C-glycosides in the existing literature, we tentatively identified compounds **3**, **4**, **5**, **6**, and **7** as luteolin-6,8-di-C-glucoside, victern-2, chrysoeriol-6,8-C-glucoside, luteolin-6-C-glucoside, and orientin, respectively.

##### Analysis of PMFs

PMFs are almost exclusively found in citrus species [40]. They have the same aglycon structure; the difference lies in the number and position of the hydroxyl groups (-OH) and/or methoxy groups (-OCH3) connected to the A, B, and C rings of the aglycon. Furthermore, the diagnostic ions of PMFs are [M + H–nCH_3_]^+^, [M + H–2CH_3_–H_2_O]^+^, [M + H–2CH_3_–CO]^+^, etc. [41]. By comparison with elution times and MS/MS data from previous reports [34,35], we screened 21 PMFs in CRPV. For example, by comparing with the reference standards, we identified compounds **35** and **40** as nobiletin and tangeretin, respectively. Nobiletin exhibited a quasi-molecular ion [M + H]^+^ at *m*/*z* of 403.13779 (C_18_H_16_O_7_) and displayed secondary fragments at *m*/*z* of 373.09137 [M + H-2CH_3_]^+^. Tangeretin exhibited a quasi-molecular ion [M + H]^+^ at *m*/*z* of 373.12783 (C_20_H_20_O_7_) and displayed secondary fragments at *m*/*z* of 328.0622 [M + H-CH_3_]^+^ and 343.08102 [M + H-2CH_3_]^+^. Compound **39** presented a quasi-molecular ion [M + H]^+^ at *m*/*z* 433.14835 (C_22_H_24_O_9_), which generated MS/MS ions at *m*/*z* of 418.12482 [M + H-CH_3_]^+^, 403.10187 [M + H-2CH_3_]^+^, and 385.09146 [M + H-2CH_3_-H_2_O]^+^. On comparing with the existing literature, we tentatively identified compound **39** as 3,5,6,7,8,3′,4′-Hexamethoxyflavone.

#### 2.4.2. Analysis of Other Compounds

Compounds **33** and **38** showed a quasi-molecular ion [M + H]^+^ at *m*/*z* of 471.20053 (C_26_H_30_O_8_) and 515.22668 (C_28_H_34_O_9_), respectively. The characteristic ions of these compounds were usually generated by the loss of H_2_O, CO, and CO_2_, and displayed secondary fragments at *m*/*z* of 425.19559 [M + H-H_2_O]^+^, 367.19043 [M + H-CH_2_O_2_-2CH_3_-H_2_O]^+^, 469.22083 [M + H-CH_2_O_2_]^+^, and 411.21735 [M + H-CH_2_O_2_-2CH_3_-H_2_O]^+^. When compared with the previous reports, compounds **33** and **38** were identified as limonin and nomilin [34,35].

### 2.5. Statistical Analysis

We applied different statistical analyses like the similarity analysis, PCA analysis, and OPLS-DA analysis, based on the FT-IR information. Similarity analysis between samples was calculated as Euclidean distance, and the final result is shown in Table 6. The larger the value, the higher the similarity between the samples, while the smaller the value, the lower the similarity between the samples. The similarity value between the GQP varieties is 1.00, and the similarity between the SHQP varieties is also 1.00. However, the similarity value between GQP and SHQP is 0.65~0.66. This means that there are large differences between GQP and SHQP.

We established a PCA model and an unsupervised pattern recognition technique with R^2^ (X) (cum) = 0.748 and Q^2^ (cum) = 0.588. The final PCA score scatter plot and bi-plot are shown in Figure 8a,b. As shown in Figure 8a, we separated the samples into distinct groups. GQP and SHQP were clustered together; as well as being visible to the naked eye in Figure 8b, this feature indicates that samples fell into two classes; different infrared peaks had different effects between GQP and SHQP. The infrared peaks numbered 7, 16, 2, 8, and 3 had a greater impact on the clustering of SHQP, while the rest had a greater impact on the clustering of GQP. A clear separation between GQP and SHQP was achieved, indicating the significant differences between these two species.

We established the OPLS-DA model by setting GQP and SHQP as groups I and II, respectively. The OPLS-DA scores plot was established with the model parameters of R^2^ (Y) = 0.999 and Q^2^ (cum) = 0.997 and is shown in Figure 8c. Therefore, we confirmed the OPLS-DA model based on the FT-IR data used to separate GQP and SHQP. Since the variable importance in the projection (VIP) value reflected the contribution of each variable to the grouping, we used VIP ≥ 1 as the threshold to filter and obtain eleven different infrared peaks related to species classification (Figure 8d). The VIP values of the infrared peaks numbered 13, 12, 11, 21, 15, 9, 17, 14, 3, and 5 were > 1, and all these were important for distinguishing between GQP and SHQP. The permutated R^2^ and Q^2^ values on the left were lower than the original point on the right (Figure 8e), thus indicating that the established OPLS-DA mode has high goodness of fit and predictability.

## 3. Materials and Methods

### 3.1. Chemicals and Materials

Hesperidin (China National Institute for Food and Drug Control), narirutin (Chengdu Keluo Biological Technology Co., Ltd., Chengdu, China), nobiletin (Chengdu Pusi Biological Technology Co., Ltd.; Chengdu, China), tangeretin (Chengdu Pusi Biological Technology Co., Ltd.; Chengdu, China), acetonitrile (Merck Co., Kenilworth, NJ, USA), methanol (Tianjin Concord, Tianjin, China), formic acid (Nanjing Chemical Reagents Co., Nanjing, China).

Twenty-two batches of CRPV samples, comprising the GQP and SHQP, from different provinces of China, were purchased from various pharmacies and markets (Figure 9, Appendix A). They were authenticated by Professor Feng Li (Shandong University of Traditional Chinese Medicine). According to morphological characteristics, all the samples were dried fruit or immature fruit skins of Citrus reticulata Blanco and its cultivars. G1–G12 were GQP, and these types of GQP were round thick slices, the surface was grayish-green or black-green, cut surface was yellow-white or light yellow-brown, densely living with most oil chambers, gas fragrance, bitter taste, and spicy. S13–S22 were SHQP, and these types of SHQP were irregular filamentous, surface grayish-green or black-green, cut surface was yellow-white or light yellow-brown, fragrance, bitter taste, and spicy.

### 3.2. Sample Preparation

Appropriate amounts of the reference standards were dissolved in methanol. The reference solutions were stored at 4 °C before use. All samples were ground into a thin powder through an 80-mesh sieve.

#### 3.2.1. Sample Preparation for Microscopic Observation

A small amount of homogenized sample powder was accurately weighed, and an appropriate amount of chloral hydrate test solution (chloral hydrate:distilled water:glycerin 50:15:10) was added to it. The mixture was heated and permeated, then covered with the cover glass to make microscopic observations.

#### 3.2.2. Sample Preparation for FT-IR

An appropriate amount of homogenized sample powder and potassium bromide powder were accurately weighed and dried separately in a constant temperature oven at 65 °C and 115 °C for 1.5 h, privately. Sample powder and potassium bromide (1:100) were mixed and ground under an infrared baking lamp, then pressed into flakes by a powder tablet press (Graseby Specac) for FT-IR analysis.

#### 3.2.3. Sample Preparation for UHPLC-Q-Exactive Orbitrap-MS

An appropriate amount of homogenized sample powder (0.2 g) was accurately weighed and ultrasonically extracted with 25 mL methanol/water (50:50, *v*/*v*) for 45 min at room temperature. The suspension was centrifuged at 5000 rpm for 5 min to remove residue. Then, the solution was filtered through a 0.22 μm filter for analysis.

### 3.3. Powder Color, Microscopic Identification, and Chemical Composition Analysis

#### 3.3.1. Power Color Determination Analysis

A high-quality COLORIMETER (NH300, ThreeNH Technology Co., Ltd., Shenzhen, China) was used to measure the powder color and obtain L* (brightness), a* (red-green), b* (yellow-blue), and the total color value E*ab, which was obtained by the following formula: E*ab = (L^2^ + a^2^ + b^2^)^1/2^. The chromaticity meter used the international universal light source D65 with a standard deviation of ∆E* ab < 0.07 (the average of 30 times of interval measurement after calibration of the standard whiteboard).

#### 3.3.2. Microscopic Observation Analysis

An Olympus BX53F microscope (Olympus Life Sciences, Tokyo, Japan) with MC50 lens (Mshot, Mingmei Technology Co., Ltd., Guangzhou, China) was used to observe the characteristic cells in CRPV. The microscope preview resolution was set as 2560 × 1944; the capture resolution was 2560 × 1944; no color enhancement (saturation default 100); no single color; no automatic white balance; color correction OFF.

#### 3.3.3. FT-IR Analysis

The infrared spectrum of CRPV was scanned using a Frontier FT-IR spectrometer (Perkin Elmer, Waltham, MA, USA) along with an FR-DTGS detector. Scanning parameters were as follows: scanning range of 4000–400 cm^−1^, spectral resolution of 4 cm^−1^, wavelength reactivity of ±0.02 cm^−1^, wavelength accuracy of ±0.1 cm^−1^. The interference of H_2_O and CO_2_ was deducted during scanning. After the processing of peak position and baseline correction, the maps and data were corrected using OMNIC 9.2 software (Thermo Nicolet Corporation, Madison, WI, USA).

#### 3.3.4. UHPLC-Q-Exactive Orbitrap-MS Analysis

Detection was performed using a Vanquish FLEX ULTRA-high-performance liquid chromatography system and a quadrupole orbital well high-resolution mass spectrometer Q-Exactive (Thermo Fisher Scientific, Waltham, MA, USA). Chromatographic separation was performed with an ACQUITY C_18_ column (2.1 mm × 100 mm, 1.8 μm, 100 Å; Phenomenex, Torrance, CA, USA) maintained at 30 °C with linear gradient elution using (A) water (0.01% formic acid) and (B) acetonitrile (0.01% formic acid) as the mobile phase. The optimized gradient elution procedure was as follows: 10 to 15% B (0–3 min), 15–37% B (3–30 min), 37–60% B (30–50 min), 60 to 85% B (50–65 min), and 85% B (75 min). The flow rate was 0.2 mL/min. The sample injection volume was 2 μL.

MS/MS identification was performed using a high-resolution mass spectrometer equipped with an electrospray ionization (ESI) source using a quadrupole tandem electrostatic field track well. Ion mode: positive and negative ion mode; auxiliary temperature: 350 °C; auxiliary gas flow rate: 10; sheath gas flow rate: 35; atomization voltage: 3.0 kV; capillary temperature: 350 °C; scan mode: full mass-DD MS2; scanning range: 100–1500. Sheath gas pressure auxiliary gas pressure: 30, 40, 50 arb. Nitrogen was used as an atomizer and auxiliary gas. Data were obtained using Thermo Scientific Xcalibur.

### 3.4. Statistical Analysis

Data obtained from the study are presented as mean ± standard deviation (SD). The one-way ANOVA and similarity analyses were carried out using SPSS 22.0 statistical software (SPSS, Inc., Chicago, IL, USA). Principal component analysis (PCA) and orthogonal partial least squares-discriminant analysis (OPLS-DA) were performed using SIMCA 14.0 software (Umetrics, Inc., San Jose, CA, USA).

## 4. Results and Discussion

In this experiment, we studied 22 batches of CRPV (12 kinds of GQP and 10 kinds of SHQP) based on their powder color, microscopic cells, FT-IR analysis, and composition analysis. The color of SHQP powder was comparatively brighter, with the color of GQP and SHQP being comparatively redder and yellower, respectively. The microscopic characteristic cells of GQP mainly comprised calcium oxalate crystallization, epidermal cells of the pulp capsule, and hesperidin crystal, while those of SHQP mainly comprised tracheal, calcium oxalate crystallization, exocarp, mesocarp parenchyma, hesperidin crystal, and stoma. FT-IR analysis showed that SHQP had a higher content of carboxylic acids and its esters than GQP, whereas GQP had a higher content of polysaccharides than SHQP.

We detected 44 main components using the UHPLC-Q-Exactive Orbitrap-MS. Among them, the flavanone aglycones, naringenin and hesperetin, were the unique components of GQP. Naringenin can reduce the phosphorylation of STAT3 in the hypothalamus by regulating adipocytokines, to achieve weight loss in obese rats and treat hypertension [42]. In vitro and in vivo experiments have confirmed that naringenin can reduce hepatic lipid accumulation and attenuate the inflammation in mice by downregulating the expression of the NLRP3 /NF-κB signaling pathway both in Kupffer cells and hepatocytes [43]. Moreover, it can also decrease urea, creatine, and uric acid levels, thereby protecting against rat liver and kidney damage [44]. Hesperetin reportedly has both neuroprotective and memory-improving effects, and it works by reducing both the inflammatory mediators’ expression and neuronal apoptosis [45,46].

Luteolin-6-C-glucoside, orientin, rhoifolin, and pilloin were unique components in SHQP. Previous literature has shown that orientin exhibits antibacterial effects and can inhibit the growth of Staphylococcus aureus [47]. Although both rhoifolin and pillion have shown anti-inflammatory effects, their mechanisms are different. Rhoifolin inhibits the secretion of inflammatory factors and inhibits the expression of IKKβ and IκBα in the NF-κB signaling pathway [48,49]. Additionally, it can repair liver and kidney damage in mice with acute inflammation, treat rheumatoid arthritis, and exert anti-pancreatic cancer effects [50]. Pilloin inhibits the production of inflammatory molecules in macrophages and downregulates inflammatory cytokines, thus showing good anti-inflammatory activity both in vitro and in vivo [51]. However, we found no pharmacological studies related to luteolin-6-C-glucoside.

GQP has been reported to be mainly used for breaking Qi and resolving stagnation, while SHQP is mainly used for regulating both the liver and Qi [28]. The above studies prove that the pharmacological activities and action mechanisms of the different components in both GQP and SHQP are different. These may be the reasons for the differences in the clinical efficacy of GQP and SHQP. This research can provide a reference for the establishment of different grade standards of CRPV or establish identification methods for Chinese patent medicines with different CRPV as raw materials, while also helping with their clinical development as a precision medicine. More studies on the pharmacological efficacy should be carried out in the future, thus helping to formulate more reasonable quality standards for CRPV and guide its clinical use as a precision drug.

## Figures and Tables

**Figure 1 molecules-27-03285-f001:**
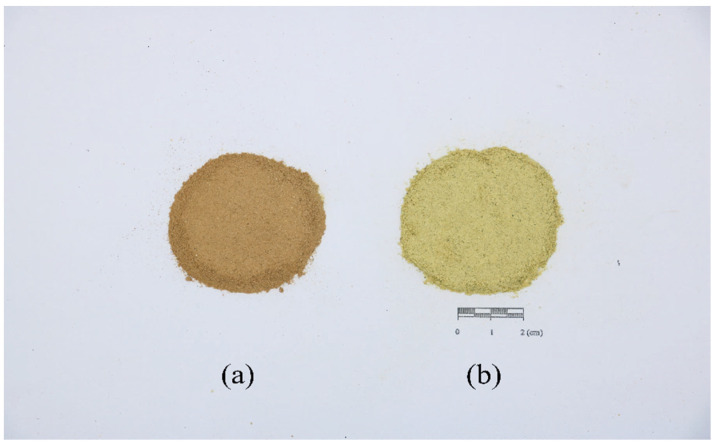
Powder color of CRPV: GQP (**a**) and SHQP (**b**).

**Figure 2 molecules-27-03285-f002:**
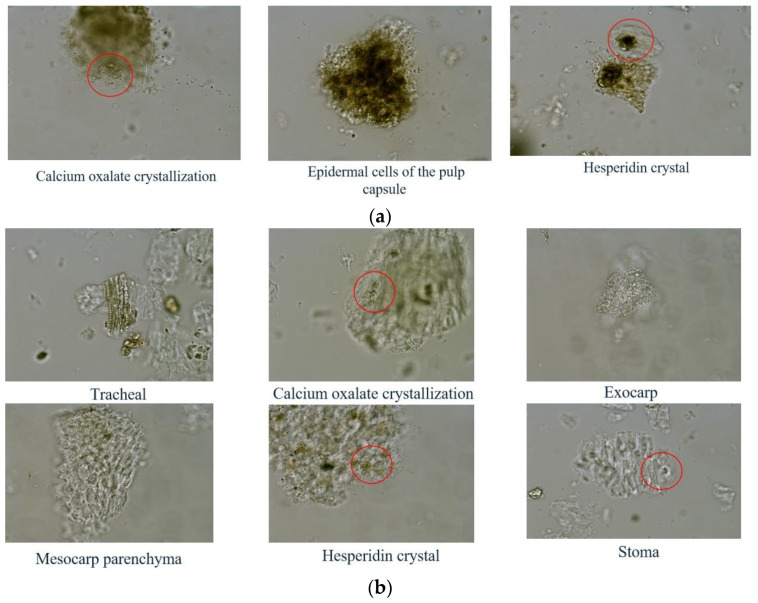
Cell images of the microscopic features of CRPV: GQP (**a**) and SHQP (**b**).

**Figure 3 molecules-27-03285-f003:**
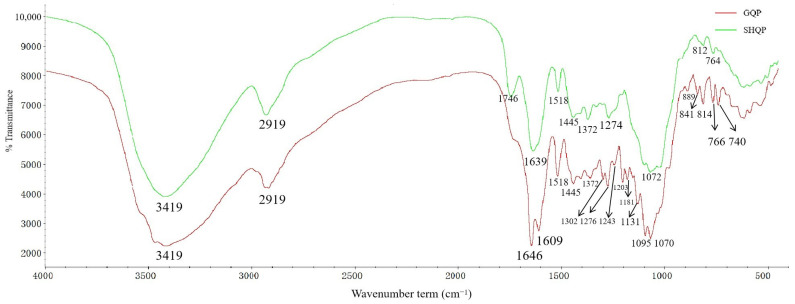
FT-IR of GQP and SHQP.

**Figure 4 molecules-27-03285-f004:**
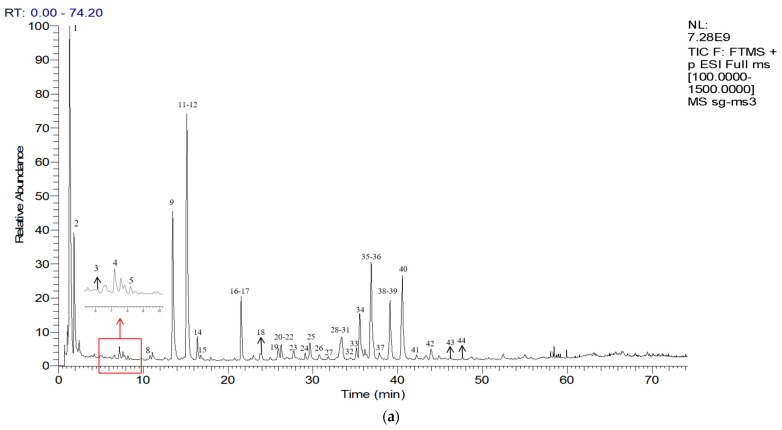
The basic peak chromatograms of CRPV in the positive ion mode and negative ion mode: GQP in positive ion mode (**a**); GQP in negative ion mode (**b**); SHQP in positive ion mode (**c**); SHQP in negative ion mode (**d**).

**Figure 5 molecules-27-03285-f005:**
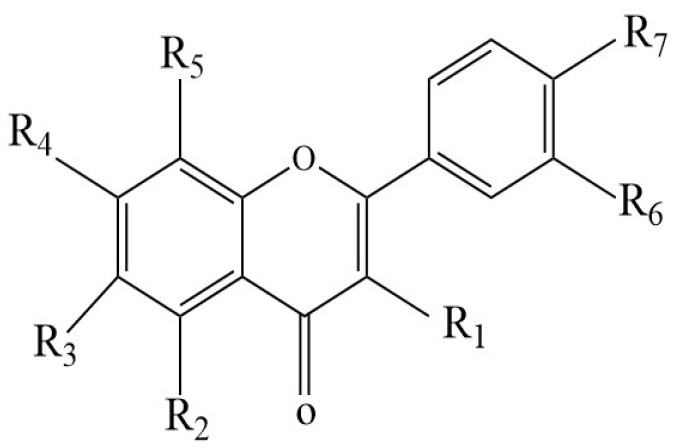
The structure of flavone.

**Figure 6 molecules-27-03285-f006:**
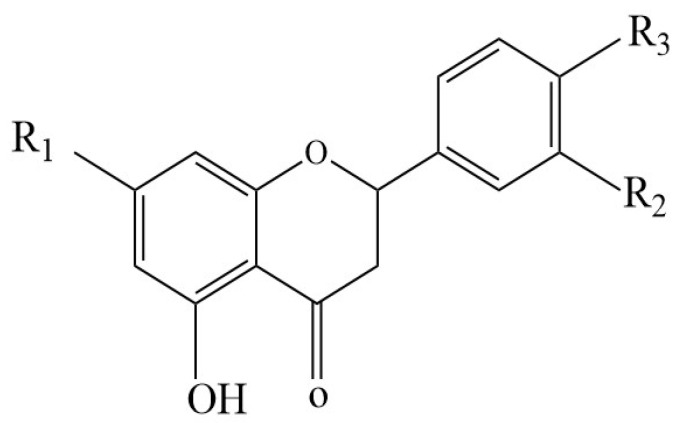
The structure of flavanone.

**Figure 7 molecules-27-03285-f007:**
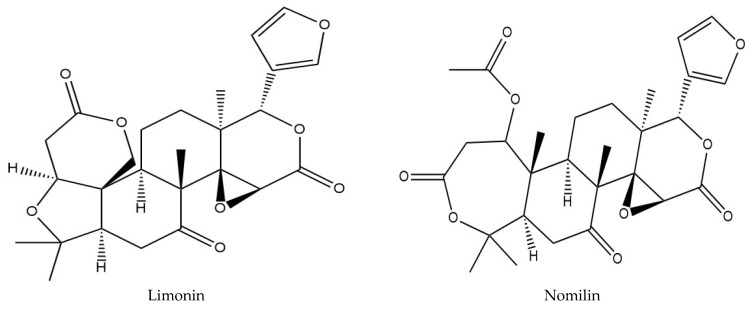
The structures of Limonin and Nomilin.

**Figure 8 molecules-27-03285-f008:**
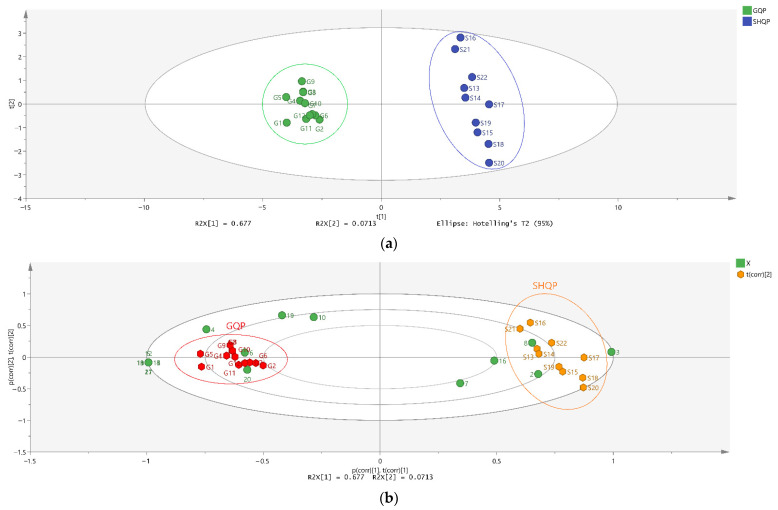
Score plot (**a**) and bi plot (**b**) of PCA model; score plot (**c**), variable importance in projection (**d**), and permutation test (**e**) of OPLS-DA model based on data of FT-IR.

**Figure 9 molecules-27-03285-f009:**
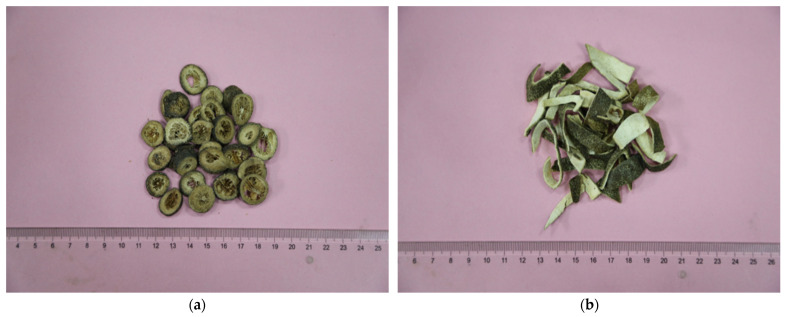
Appearance characteristics of GQP (**a**) and SHQP (**b**).

**Table 1 molecules-27-03285-t001:** The value of L*, a*, and b* of GQP and SHQP.

Variable Name	Sample Name	
	GQP (n = 12)	SHQP (n = 10)
L*	62.0 ± 3.9	72.9 ± 2.7 ^▲▲^
a*	6.0 ± 0.6	1.6 ± 1.3 ^▲▲^
b*	13.8 ± 1.7	19.1 ± 1.7 ^▲▲^
E*ab	63.8 ± 4.1	75.6 ± 2.5 ^▲▲^

^▲▲^*p* < 0.01, as compared with the GQP.

**Table 2 molecules-27-03285-t002:** Location of the most relevant features of the FT-IR spectra of the CRPV.

Name	Wavenumber Term (cm^−1^)																				
	1	2	3	4	5	6	7	8	9	10	11	12	13	14	15	16	17	18	19	20	21
GQP	3419	2919	—	1646	1609	1518	1445	1372	1302	1276	1243	1203	1181	1131	1095	1070	889	841	814	766	740
SHQP	3419	2919	1746	1639	—	1518	1445	1372	—	1274	—	—	—	—	—	1072	—	—	812	764	—

**Table 3 molecules-27-03285-t003:** Identification of the chemical compounds of CRPV by UHPLC-Q-Exactive Orbitrap-MS.

Compound/Peak Number	Identification	Molecular Formula	Retention Time (min)	[M + H]^+^ (*m*/*z*) (Error, ppm)	[M − H]^−^(*m*/*z*) (Error, ppm)	Fragment Ions in the Positive Ion Mode (*m*/*z*) ^b^	Fragment Ions in the Negative Ion Mode (*m*/*z*) ^b^	Belongs to Medicine
**Flavone-C-glycosides**								
3	Luteolin-6,8-di-C-glucoside ^c^	C_27_H_30_O_16_	6.12	611.15961 (−1.72)	ND	557.12787 [M + H-OCH_2_CH_3_]^+^, 473.10663 [M + H-C_4_H_8_O_4_-H_2_O]^+^, 353.06503 [M + H-2C_4_H_8_O_4_-H_2_O]^+^	ND	G/S
4	Victern-2 (apigenin-6,8-di-C-glucoside) ^c^	C_27_H_30_O_15_	7.21	595.16406 (−2.834)	593.15009 (−0.011)	457.11102 [M + H-C_4_H_8_O_4_-H2O]^+^	ND	G/S
5	Chrysoeriol-6,8-C-glucoside ^c^	C_28_H_22_O_16_	7.72	625.17480 (−2.417)	623.16058 (−0.13)	487.11911 [M + H-C_4_H_8_O_4_-H_2_O]^+^, 367.08112 [M + H-2C_4_H_8_O_4_-H_2_O]^+^	503.12042 [M − H-C_4_H_8_O_4_]^−^, 413.08749 [M − H-C_4_H_8_O_4_-C_3_H_6_O_3_]^−^, 383.07684 [M − H-2C_4_H_8_O_4_]^−^	G/S
6	Luteolin-6-C-glucoside ^c^	C_21_H_20_O_11_	9.04	449.10727 (−1.264)	447.09268 (1.101)	413.08545 [M + H-2H_2_O]^+^, 395.07538 [M + H-3H_2_O]^+^, 329.06519 [M + H-C_4_H_8_O_4_]^+^, 299.05475 [M + H-C_8_H_6_O_3_]^+^	ND	S
7	Orientin ^c^	C_21_H_20_O_11_	9.52	449.10687 (−2.155)	ND	431.09659 [M + H-H_2_O]^+^, 413.08609 [M + H-2H_2_O]^+^, 383.07559, 353.06506, 329.06506, 311.05478, 299.05450	ND	S
**Flavone-O-glycosides**								
8	Eriodictyol-7-O-rutinoside ^c^	C_27_H_32_O_15_	10.83	597.17963 (−1.767)	597.17963 (−1.767)	289.07040, 355.08051, 195.02887, 153.01813	ND	G/S
10	Rhoifolin ^c^	C_27_H_30_O_14_	14.01	579.16992 (−1.575)	577.147 (0.257)	433.11099 [M + H-Rha]^+^, 271.05981 [M + H-Glc-Rha]^+^, 153.01849, 85.02903	269.04529 [M − H-Glc-Rha]^−^	S
14	3-Hydroxy-3-methyl-5-oxo-5-[[(2R,3S,4S,5R,6S)-3,4,5-trihydroxy-6-[2-methoxy-4-(5,7,8-trihydroxy-6-methoxy-4-oxochromen-2-yl)phenoxy]oxan-2-yl]methoxy]pentanoic acid/2-(3-Methoxy-4-hydroxyphenyl)-3-[6-O-(3,5-dihydroxy-3-methyl-5-oxopentanoyl)-beta-D-glucopyranosyloxy]-5,7-dihydroxy-8-methoxy-4H-1-benzopyran-4-one ^c^	C_29_H_32_O_17_	16.44	653.16907 (−2.156)	651.15558 (0.007)	347.07596, 332.05252, 301.03387, 187.05991, 145.04945, 127.03915, 85.02909	286.04791, 151.00246	G/S
15	Torminaloside ^c^	C_28_H_30_O_16_	16.84	623.15985 (−1.302)	621.64514 (−0.208)	317.06540	315.06540 [M − H-144-162]^−^, 271.02466, 151.00233	G/S
**Flavanone-O-glycosides**								
9	Narirutin ^a,c^	C_27_H_32_O_14_	13.51	581.18439 (−2.092)	579.17035 (−0.832)	273.07533 [M + H-Glc-Rha]^+^, 195.02816, 153.01816, 85.02905	271.06091 [M − H-Glc-Rha]^−^, 151.00253	G/S
12	Hesperidin ^a,c^	C_28_H_34_O_15_	15.18	611.19501 (−2.037)	609.18109 (−0.503)	303.08820 [M + H-Glc-Rha]^+^, 263.05466, 219.02869, 111.04427, 85.02897	301.07141 [M − H-Glc-Rha]^−^,286.04776	G/S
16	Didymin ^c^	C_28_H_34_O_14_	21.57	595.20013 (−3.364)	593.18616 (−0.543)	287.09140, 153.0204 [M + H-Glc-Rha-C _9_H_10_O]^+^, 129.05472, 111.04424, 85.02905	285.07639 [M − H-Glc-Rha]^−^	G/S
20	Melitidin ^c^	C_33_H_40_O_18_	26.34	725.22607 (−3.682)	723.21222 (−0.788)	419.13303	ND	G/S
**Flavanone aglycones**								
11	Homoeriodictyol ^c^	C_16_H_14_O_6_	15.19	303.08502 (−4.472)	301.07068 (0.051)	153.01822 [M + H-C_9_H_10_O_2_]^+^	286.04791 [M − H-CH_3_]^−^,151.00233 [M − H-C_9_H_10_O_2_]^−^	G/S
17	Isosakuranetin ^c^	C_16_H_14_O_5_	21.57	287.09030 (−3.832)	285.07590 (0.526)	153.01819 [M + H-C_9_H_10_O]^+^, 133.06485, 161.05966	243.06580 [M − H-C_2_H_2_O]^−^, 151.00258 [M − H-C_9_H_10_O]^−^	G/S
19	Naringenin ^c^	C_15_H_12_O_5_	25.93	273.07462 (−2.380)	271.06021 (0.406)	153.01817 [M + H-C_8_H_8_O]^+^, 171.02881, 147.03499, 119.04930, 91.05465, 67.01865	151.00258 [M − H-C_8_H_8_O]^−^, 119.04896 [M − H-C_7_H_4_O_4_]^−^, 107.01255 [M − H-C_8_H_8_O-CO_2_]^−^	G
23	Hesperetin ^c^	C_16_H_14_O_6_	27.73	303.08527 (−3.447)	301.07068 (0.051)	153.01822 [M + H-C_9_H_10_O_2_]^+^, 287.07544, 219.06482, 171.02875, 177.05461	ND	G
**PMFs**								
13	Pilloin ^c^	C_17_H_14_O_6_	15.60	315.08542 (−0.895)	ND	300.06253 [M + H-CH_3_]^+^, 285.03963 [M + H-2CH_3_]^+^, 136.01561 [M + H-CH_3_-C_9_H_8_O_3_]^+^	ND	S
21	Monohydroxy-trimethoxyflavone ^c^	C_18_H_16_O_6_	26.55	329.10101 (−2.901)	ND	314.07339 [M + H-CH_3_]^+^, 299.05457 [M + H-2CH_3_]^+^, 268.07254, 181.01285, 153.01811	ND	G/S
22	7-Hydroxy-3,5,6,8-tetramethoxyflavone ^c^	C_19_H_18_O_7_	26.59	359.11145 (−3.006)	ND	344.08826 [M + H-CH_3_]^+^, 301.07016 [M + H-2CH_3_-CO]^+^, 298.08319	ND	G/S
24	7-Hydroxy-3′,4′,5,6,8-pentamethoxyflavone ^c^	C_20_H_20_O_8_	29.10	389.12201 (−2.786)	ND	374.09921 [M + H-CH_3_]^+^, 359.07571 [M + H-2CH_3_]^+^, 341.06537 [M + H-2CH_3_-H_2_O]^+^	ND	G/S
25	Isosinensetin (3′,4′,5,7,8-Pentamethoxyflavone) ^c^	C_20_H_20_O_7_	29.66	373.12717 (−2.705)	ND	343.08087 [M + H-2CH_3_]^+^, 315.08609 [M + H-2CH_3_-CO]^+^	ND	G/S
26	3′-Hydroxy-4′,5,6,7,8-Pentramethoxyflavone ^c^	C_20_H_20_O_8_	30.77	389.12219 (−2.786)	ND	374.09912 [M + H-CH_3_]^+^, 359.07562 [M + H-2CH_3_]^+^	ND	G/S
27	3,4′,5,7-Tetrametho-xyflavone ^c^	C_19_H_18_O_6_	31.86	343.11694 (−1.967)	ND	327.08597 [M + H-CH_4_]^+^, 328.09256 [M + H-CH_3_]^+^, 299.09109 [M + H-CH_4_-CO]^+^, 163.03397	ND	G/S
28	Monohydroxy-hexamethoxyflavone ^c^	C_21_H_22_O_9_	32.74	419.13324 (−0.999)	ND	404.10974 [M + H-CH_3_]^+^, 389.08646 [M + H-2CH_3_]^+^	ND	G/S
29	5-Hydroxy-3′,4′,7,8-tetramethoxyflavone ^c^	C_19_H_18_O_7_	33.12	359.11148 (−2.922)	ND	197.0105 [M + H-C_10_H_10_O_2_]^+^, 169.0165 [M + H-C_10_H_10_O_2_-CO]^+^	ND	G/S
30	Sinensetin (3′,4′,5,6,7-Pentrametho-xyflavone) ^c^	C_20_H_20_O_7_	33.22	373.12723 (−2.545)	ND	357.09563 [M + H-CH_4_]^+^, 343.08081 [M + H-2CH_3_]^+^	ND	G/S
31	4′,5,6,7-Tetramethoxyflavone	C_19_H_18_O_6_	33.43	343.11673 (−2.579)	ND	328.09363 [M + H-CH_3_]^+^, 317.07031, 285.07556, 181.01324, 153.01805	ND	G/S
32	3′,4′,5,7,8-Pentamethoxyflavanone ^c^	C_20_H_22_O_7_	34.47	375.14322 (−0.609)	ND	211.05995 [M + H-C_10_H_10_O_2_]^+^, 196.03661 [M + H-C_10_H_10_O_2_-CH_3_]^+^, 168.04150	ND	G/S
35	Nobiletin (3′,4′,5,6,7,8-hexamethoxyflavone) ^a,c^	C_21_H_22_O_8_	36.90	403.13779 (−2.367)	ND	373.09137 [M + H-2CH_3_]^+^, 355.08060 [M + H-3CH_3_]^+^, 327.08591, 312.0705, 301.07031, 211.02361, 183.02884	ND	G/S
36	4′,5,7,8-Teramethoxyflavone ^c^	C_19_H_18_O_6_	37.23	343.11685 (−2.229)	ND	313.07028 [M + H-2CH_3_]^+^, 327.08560 [M + H-CH_4_]^+^	ND	G/S
37	Dihydroxy-tetramethoxyflavone ^c^	C_19_H_18_O_8_	37.86	375.10706 (−1.024)	ND	360.08340 [M + H-CH_3_]^+^, 345.06018 [M + H-2CH_3_]^+^, 327.04971 [M + H-2CH_3_-H_2_O]^+^	ND	G/S
39	3,5,6,7,8,3′,4′-Hexamethoxyflavone ^c^	C_22_H_24_O_9_	39.14	433.14835 (−2.213)	ND	418.12482 [M + H-CH_3_]^+^,403.10187 [M + H-2CH_3_]^+^, 385.09146 [M + H-2CH_3_-H_2_O]^+^, 345.05917 [M + H-4CH_3_-CO]^+^, 211.02190, 165.05423	ND	G/S
40	Tangeretin (4′,5,6,7,8-Pentamethoxyflavone) ^a,c^	C_20_H_20_O_7_	40.56	373.12738 (−2.143)	ND	358.10428, 343.08102 [M + H-2CH_3_]^+^, 297.07553, 183.02886, 135.04411	ND	G/S
41	Monohydroxy-tetramethoxyflavone ^c^	C_19_H_18_O_7_	42.31	359.11206 (−1.307)	ND	343.08096, 315.08582, 298.08322, 164.08301	ND	G/S
42	5-Hydroxy-6,7,8,3′,4′-Pentamethoxyflavone ^c^	C_20_H_20_O_8_	43.91	389.12213 (−2.477)	ND	374.09550, 359.07574 [M + H-2CH_3_]^+^, 341.06528, 215.01813, 197.0119 [M + H-C_10_H_12_O_2_-2CH_3_]^+^	ND	G/S
43	Natsudaidain (3-hydroxy-3′,4′,5,6,7,8-Hexamethoxyflavone) ^c^	C_21_H_22_O_9_	46.20	419.13290 (−1.810)	ND	389.08630 [M + H-2CH_3_]^+^, 371.07581 [M + H-2CH_3_-H_2_O]^+^, 361.09167	ND	G/S
44	Monohydroxy-tetramethoxyflavone ^c^	C_19_H_18_O_7_	47.65	359.11179 (−2.0598)	ND	344.08826 [M + H-CH_3_]^+^, 329.06528 [M + H-2CH_3_]^+^, 197.00768	ND	G/S
**Alkaloids**								
1	Stachydrine ^c^	C_7_H_13_NO_2_	1.30	144.10161	ND	102.05531, 84.08133, 58.06592	ND	G/S
2	Synephrine ^a,c^	C_9_H_13_NO_2_	1.84	168.10179	ND	135.06801 [M + H-H_2_O-CH3]^+^, 119.04944 [M + H-H_2_O-CH3NO2]^+^, 107.04957 [M + H-H_2_O-CH_3_-CO]^+^, 91.05482, 81.07052	ND	G/S
18	CitrusinIII ^c^	C_36_H_53_N_7_O_9_	23.97	728.39557 (−2.722)	ND	700.40399 [M + H-CO]^+^, 587.31830, 474.23471	ND	G/S
**Limonoids**								
33	Limonin ^c^	C_26_H_30_O_8_	35.15	471.20053 (−1.728)	469.18555 (−0.307)	425.19559 [M + H-H_2_O]^+^, 367.19043 [M + H-CH_2_O_2_-2CH_3_-H_2_O]^+^, 161.05972	ND	G/S
38	Nomilin ^c^	C_28_H_34_O_9_	39.03	515.22668 (−1.706)	ND	469.22083 [M + H-CH_2_O_2_]^+^, 411.21735 [M + H-CH_2_O_2_-2CH_3_-H_2_O]^+^, 161.05966	ND	G/S
**Other compounds**								
14	2-(carbamoylamino)ethyl-[5-(dimethylamino)pentyl]-methyl-propylazanium/diethyl-[[[2-methyl-3-[1-(methylamino)butan-2-ylamino]propanoyl]amino]methyl]azanium ^c^	C_14_H_33_ON_4_	35.54	274.27328 (−2.066)	ND	256.26324, 106.08664, 88.07628	ND	G/S

G: Geqingpi belongs to CRPV; S: Sihuaqingpi belongs to CRPV; ^a^ Confirmation in comparison with authentic standards.; ^b^ Glc = glucose moiety, Rha = rhamnose moiety, ND: Not detected; ^c^ Confirmation in comparison with literature.

**Table 4 molecules-27-03285-t004:** Compounds with the flavone structure.

Name	R1	R2	R3	R4	R5	R6	R7
Luteolin-6,8-di-C-glucoside	H	OH	Glc	OH	Glc	OH	OH
Victern-2 (apigenin-6,8-di-C-glucoside)	H	OH	Glc	OH	Glc	H	OH
Chrysoeriol-6,8-C-glucoside	H	OH	Glc	OH	Glc	OCH_3_	OH
Rhoifolin	H	OH	H	ONeo	H	H	OH
Isosinensetin (3′,4′,5,7,8-Pentamethoxyflavone)	OCH_3_	H	OCH_3_	OCH_3_	OCH_3_	OCH_3_	OCH_3_
Sinensetin (5,6,7,3′,4′-Pentramethoxyflavone)	OCH_3_	OCH_3_	OCH_3_	H	OCH_3_	OCH_3_	OCH_3_
Nobiletin (5,6,7,8,3′,4′-hexamethoxyflavone)	OCH_3_	OCH_3_	OCH_3_	OCH_3_	OCH_3_	OCH_3_	OCH_3_
Tangeretin (5,6,7,8,4′-Pentamethoxyflavone)	OCH_3_	OCH_3_	OCH_3_	OCH_3_	H	H	OCH_3_
3,5,6,7,8,3′,4′-Hexamethoxyflavone	H	OCH_3_	OCH_3_	OCH_3_	OCH_3_	OCH_3_	OCH_3_

Glc = glucose, Neo = neohesperidoside.

**Table 5 molecules-27-03285-t005:** Compounds with the flavanone structure.

Name	R1	R2	R3
Narirutin	ORut	H	OH
Hesperidin	ORut	OH	OCH_3_
Homoeriodictyol	OH	OCH_3_	OH
Isosakuranetin	OH	H	OCH_3_
Didymin	ONeo	H	OCH_3_
Naringenin	OH	H	OH
Hesperetin	OH	OH	OCH_3_

Rut = rutinoside, Neo = neohesperidoside.

**Table 6 molecules-27-03285-t006:** Similarity analysis of GQP and SHQP.

Sample Number	G1	G2	G3	G4	G5	G6	G7	G8	G9	G10	G11	G12	S13	S14	S15	S16	S17	S18	S19	S20	S21	S22
G1	1.00																					
G2	1.00	1.00																				
G3	1.00	1.00	1.00																			
G4	1.00	1.00	1.00	1.00																		
G5	1.00	1.00	1.00	1.00	1.00																	
G6	1.00	1.00	1.00	1.00	1.00	1.00																
G7	1.00	1.00	1.00	1.00	1.00	1.00	1.00															
G8	1.00	1.00	1.00	1.00	1.00	1.00	1.00	1.00														
G9	1.00	1.00	1.00	1.00	1.00	1.00	1.00	1.00	1.00													
G10	1.00	1.00	1.00	1.00	1.00	1.00	1.00	1.00	1.00	1.00												
G11	1.00	1.00	1.00	1.00	1.00	1.00	1.00	1.00	1.00	1.00	1.00											
G12	1.00	1.00	1.00	1.00	1.00	1.00	1.00	1.00	1.00	1.00	1.00	1.00										
S13	0.66	0.66	0.65	0.65	0.65	0.66	0.65	0.65	0.65	0.65	0.65	0.65	1.00									
S14	0.65	0.65	0.65	0.65	0.65	0.65	0.65	0.65	0.65	0.65	0.65	0.65	1.00	1.00								
S15	0.66	0.65	0.65	0.65	0.65	0.65	0.65	0.65	0.65	0.65	0.65	0.65	1.00	1.00	1.00							
S16	0.66	0.66	0.65	0.65	0.65	0.65	0.65	0.65	0.65	0.65	0.65	0.65	1.00	1.00	1.00	1.00						
S17	0.65	0.65	0.65	0.65	0.65	0.65	0.65	0.65	0.65	0.65	0.65	0.65	1.00	1.00	1.00	1.00	1.00					
S18	0.66	0.66	0.66	0.66	0.66	0.66	0.66	0.65	0.66	0.66	0.66	0.66	1.00	1.00	1.00	1.00	1.00	1.00				
S19	0.65	0.65	0.65	0.65	0.65	0.65	0.65	0.65	0.65	0.65	0.65	0.65	1.00	1.00	1.00	1.00	1.00	1.00	1.00			
S20	0.66	0.66	0.65	0.65	0.66	0.66	0.65	0.65	0.66	0.66	0.65	0.65	1.00	1.00	1.00	1.00	1.00	1.00	1.00	1.00		
S21	0.65	0.65	0.65	0.65	0.65	0.65	0.65	0.65	0.65	0.65	0.65	0.65	1.00	1.00	1.00	1.00	1.00	1.00	1.00	1.00	1.00	
S22	0.66	0.65	0.65	0.65	0.65	0.65	0.65	0.65	0.65	0.65	0.65	0.65	1.00	1.00	1.00	1.00	1.00	1.00	1.00	1.00	1.00	1.00

## Data Availability

Data is contained within the article or Appendix A.

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
