# Peer review of "Analyses of Physical and Chemical Compositions of Different Medicinal Specifications of CRPV by Use of Multiple Instrumental Techniques Combined with Multivariate Statistical Analysis"

_molecules, 2022, doi:10.3390/molecules27103285_

Round 1

Reviewer 1 Report

First of all, I would like to thank the editor for letting me to review this manuscript.

The authors present a work in which two different products derived from Citri Reticulatae Pericarpium Viride i.e., GQP and SHQP, are compared using various physicochemical parameters. The colour of the preparation, microscopic characteristics, FT-IR spectra and the determination of, mainly flavonoid related compounds by HPLC-MS are the parameters used to carry out the comparison.

The results show differences in colour, the characteristics observed under the microscope, some features of the FT-IR spectra and some of the compounds identified by HPLC-MS.

The main objection to the manuscript is that the novelty and usefulness of the manuscript are not clearly stated. In the literature, as indicated by the authors, there are several published works that analyzed samples of Citri Reticulatae Pericarpium Viride, although they do not focus on the differences of CRPV at different stages of maturity. It seems clear that the composition of a plant part will be different depending on its maturity, therefore, in my opinion, the interest of the work should not be to highlight these differences but to apply these differences to some objective (medical, quality control, fraud ...). Therefore, the novel contribution and the usefulness of this work should be presented more clearly so potential readers can get its importance easily.

Other specific comments are:

Throughout the text acronyms or abbreviations are used, it is important that all of them are defined in the document the first time they appear.

The quality of the pictures in Figure 1 should be improved to better appreciate the characteristics of the two preparations.

The data presented in Table 1 should have the appropriate number of significant figures. 

In my opinion heading of Table 2 could be more descriptive since the data does not identify the peaks of the spectra but their location, i.e. the wavenumber. In addition wavenumber term is commonly used instead of wave number.

Figures 3 and 4 should be made larger for better readability.

Table 4. How has similarity been calculated? Which index was used and why was it selected.

Figure 6 presents the data of the multivariate statistical analysis, however, no effort is made to explain which variables (characteristics of the spectra) are the ones that create the groups of samples, i.e., what spectra features are the most useful for characterizing an unknown sample?

Reviewer 2 Report

The MS entitled: "Analyze the Characteristics of Different Medicinal Specifica-2 tions of CRPV by Multiple Analysis Techniques combined 3 with Multivariate Statistics" by Wang et al is a comparative study of two specifications of CRPV that are extensively used in herbal Chinese Medicine. The study compares the physical and chemical composition of GQP and SHQP in terms of color and chemical composition. The authors employed various techniques including microscopic cellular examination, FT-IR spectroscopy, as well as Orbitrap MS. The topic in itself is not of high novelty as numerous papers have described the chemical composition of CRPV. However, the fact that the authors put much effort in comparing the physical and chemical characteristics of the two commonly utilized medicinal formulations of CRPV, namely GQP and SHQP this is rather novel and of great importance.  Despite the fact that this approahc has been recently reported by He et al (2021) in their paper:

He, X.; Huang, S.; Wu, M.; Wu, M.; Zhang, Y.; Ma, Z.; Liu, L.; Cao, H. Simultaneous quantitative analysis of ten bioactive 233 flavonoids in Citri Reticulatae Pericarpium Viride (Qing Pi) by ultrahigh‐performance liquid chromatography and high‐reso-234 lution mass spectrometry combined with chemometric methods. Phytochem. Analysis 2021, 32, 1152-1161.

The MS has merit as it determines the differences with employing a rather different approaches. However, the MS in its present form suffers from major flaws that authors need to correct before consideration of its publication. A detailed account of my comments is listed below:

MAJOR:

  • The major CRPV components are volatile oils and flavonoids. Previous studies employing two-dimensional gas chromatography with high-resolution time-of-flight mass spectrometry (GC × GC-HR-TOFMS) identified a total of 176 compounents majorely represented by falvonoids. However the current study employging Orbitrap MS identified only 44. This clearly show that this instrumental methodology is not ideal for these types of essential oil-rich plant samples. Authors need to comment on this both in the introduction art as well as discuss these differences. For example, β-Elemene is a triterpinoid found in CRPV with potent antineoplastic activity and enhancing potential for the cytotoxicity of various chemotherapeutic drugs as well as reduces the accompanied side effects of chemotherapy. Authors are advised to consult the following reference:

Qin K, Zheng L, Cai H, Cao G, Lou Y, Lu T, Shu Y, Zhou W, Cai B. Characterization of Chemical Composition of Pericarpium Citri Reticulatae Volatile Oil by Comprehensive Two-Dimensional Gas Chromatography with High-Resolution Time-of-Flight Mass Spectrometry. Evid Based Complement Alternat Med. 2013;2013:237541. doi: 10.1155/2013/237541. Epub 2013 Apr 28. PMID: 23710215; PMCID: PMC3655619.

  • The MS despite its originality in comparing two different specifications of CRPV (GQP and SHQP), cannot be published in its present from as its English qualities is rather weak and suffers from serious and frequent grammatical mistakes and syntax errors. The MS should benefit from the help of proffessional English editing service or alternatively from a native speaker.

MINOR

TITLE: The current title is misleading and confusing as it contains serious grammatical and syntax errors. I suggest changing it to the following:

"Analyses of physical and chemical composition of different medicinal specifications of CRPV by use of multiple instrumental techniques combined with multivariate statistical analysis"

ABSTRACT

-The chemical composition analysis of CRPV by use of Ultra-high-resolution mass spectrometry Q-Exactive™ HF Hybrid Quadrupole-Orbitrap™ coupled with liquid chromatography (UHPLC) must be inserted in full in the abstract. It's the main state-of-the-art technology used for the identifications of phytochemicals.

-A concluding statement in the abstract is missing describing the significance of the findings particularly in terms of differences in flavonoid contents found between GQP and SHQP. How would that benefit the scientific readers and indeed what implications would these findings have to help consumers prefer one formulation from the other. 

INTRODUCTION:

-There is a serious lack of a paragraph elaborating on the medicinal benefits of CRPV as a natural therapeutic agent for the treatment of numerous diseases such as inflammatory syndromes of the respiratory tract including bronchitis and asthma, liver injury and cancers (liver, colon, etc.). To list a few, the authors might benefit from the following papers:

-Kang SA, Park HJ, Kim MJ, Lee SY, Han SW, Leem KH. Citri Reticulatae Viride Pericarpium extract induced apoptosis in SNU-C4, human colon cancer cells. J Ethnopharmacol. 2005 Feb 28;97(2):231-5. doi: 10.1016/j.jep.2004.11.012. Epub 2005 Jan 11. PMID: 15707758.

-Wu J, Ye X, Yang S, Yu H, Zhong L, Gong Q. Systems Pharmacology Study of the Anti-Liver Injury Mechanism of Citri Reticulatae Pericarpium. Front Pharmacol. 2021 Apr 12;12:618846. doi: 10.3389/fphar.2021.618846. PMID: 33912040; PMCID: PMC8072898.

RESULTS

This section needs to be changed to "Results and discussion"

-Authors are required to discuss in details the implication of the findings on the therapeutic potential of each specification GQP and SHQP. More elaboration on the usefulness of these findings from an applied pharmaceutical and/or ethnomedicinal angle, supported with recent literatue. This may be conducted in a final paragraph just before the conslusion.

Round 2

Reviewer 1 Report

Thank you for the effort made to take into account the comments of the first review.  Some of these items have been satisfactorily resolved, however other items need some more work.

Point 4. In my view, the data in Table 1 are not presented correctly from a statistical point of view. For example, a value of 62.011 shows many figures that do not make sense considering the precision of the measurement method (+-3.875). If this last number is the standard deviation it should be shown as +-3.9 which would force to present the mean value as 62.0.

Point 5. The heading of Table 5 has been modified, however, I believe it should be worded similarly to "Location of the most relevant features of the FTIR spectra of the samples"

Point 7. I am not able to locate in the text the type of similarity index that has been calculated to elaborate Table 4, i.e, Euclidean, Mahalanobis, Manhattan....? Please specify.

Reviewer 2 Report

I'm now satisfied with the revised version of the MS. No further comments are needed. 
